# The Perpetual Vector Mosquito Threat and Its Eco-Friendly Nemeses

**DOI:** 10.3390/biology13030182

**Published:** 2024-03-12

**Authors:** Leticia Silva Miranda, Sarah Renee Rudd, Oscar Mena, Piper Eden Hudspeth, José E. Barboza-Corona, Hyun-Woo Park, Dennis Ken Bideshi

**Affiliations:** 1Graduate Program in Biomedical Sciences, Department of Biological Sciences, California Baptist University, Riverside, CA 92504, USA; leticiasilva.miranda@calbaptist.edu (L.S.M.); or srudd@students.llu.edu (S.R.R.); hpark@calbaptist.edu (H.-W.P.); 2Integrated Biomedical Graduate Studies, and School of Medicine, Loma Linda University, Loma Linda, CA 92350, USA; 3Undergraduate Program in Biomedical Sciences, Department of Biological Sciences, California Baptist University, Riverside, CA 92504, USA; omena@calbaptist.edu (O.M.); pipereden.hudspeth@calbaptist.edu (P.E.H.); 4Departmento de Alimentos, Posgrado en Biociencias, Universidad de Guanajuato Campus Irapuato-Salamanca, Irapuato 36500, Guanajuato, Mexico; josebar@ugto.mx

**Keywords:** vector mosquitoes, *Aedes*, *Anopheles*, *Culex*, arboviruses, climate change, biocontrol, green technology, *Bacillus thuringiensis* subsp. *israelensis*, *Lysinibacillus sphaericus*, prokaryotic insect larvicidal organelle (PILO), SIT, RIDL, IIT, *Wolbachia*, CRIPSR/Cas9, CRISPR/Cas13

## Abstract

**Simple Summary:**

*Aedes*, *Culex*, and *Anopheles* mosquitoes are the most prolific arthropod vectors of viral and parasitic agents of debilitating and lethal diseases in humans and animals. Despite some success in integrated pest management programs to control vectors, mosquito-borne diseases, such as dengue and dengue hemorrhagic fever, yellow fever, chikungunya, West Nile, and Zika, and parasitic diseases, such as malaria, lymphatic filariasis, and river blindness, continue to threaten the health and well-being of half the world’s population, many of whom live in economically and medically challenged societies. The perpetual problem inflicted by vector-borne diseases is compounded by the selection for resistance to synthetic pesticides, globalization, and climate change. The latter appears to be the most significant factor implicated in the geographic expansion of mosquitoes. Here, we present a review of these challenges and highlight traditional vector control strategies that employ synthetic pesticides, and “green” eco-friendly technologies that include SIT, IIT, RIDL, CRISPR/Cas9/Cas13 gene drive systems, and biological control, with an emphasis on *Lysinibacillus sphaericus* and *Bacillus thuringiensis* subsp. *israelensis* (Bti).

**Abstract:**

Mosquitoes are the most notorious arthropod vectors of viral and parasitic diseases for which approximately half the world’s population, ~4,000,000,000, is at risk. Integrated pest management programs (IPMPs) have achieved some success in mitigating the regional transmission and persistence of these diseases. However, as many vector-borne diseases remain pervasive, it is obvious that IPMP successes have not been absolute in eradicating the threat imposed by mosquitoes. Moreover, the expanding mosquito geographic ranges caused by factors related to climate change and globalization (travel, trade, and migration), and the evolution of resistance to synthetic pesticides, present ongoing challenges to reducing or eliminating the local and global burden of these diseases, especially in economically and medically disadvantaged societies. Abatement strategies include the control of vector populations with synthetic pesticides and eco-friendly technologies. These “green” technologies include SIT, IIT, RIDL, CRISPR/Cas9 gene drive, and biological control that specifically targets the aquatic larval stages of mosquitoes. Regarding the latter, the most effective continues to be the widespread use of *Lysinibacillus sphaericus* (Ls) and *Bacillus thuringiensis* subsp. *israelensis* (Bti). Here, we present a review of the health issues elicited by vector mosquitoes, control strategies, and lastly, focus on the biology of Ls and Bti, with an emphasis on the latter, to which no resistance has been observed in the field.

## 1. Introduction

Vectors are organisms capable of transmitting debilitating and deadly pathogens to humans and animals. The World Health Organization (WHO) estimates that each year roughly 700,000–1,000,000 people succumb to vector-borne diseases [1]. Arthropod vectors include mosquitoes, black flies, tsetse flies, sand flies, triatome bug, fleas, and ticks that collectively transmit viral, parasitic, and bacterial pathogens. Well-known examples of vector-borne diseases are (i) chikungunya, dengue and severe dengue, yellow fever, Zika, and West Nile (viruses); (ii) malaria, leishmaniasis, Chagas disease, lymphatic filariasis, and onchocerciasis (protozoal and helminth parasites); and (iii) Lyme disease, tularemia, plague, and rickettsia-type diseases, including typhus and Rocky Mountain spotted fever, Q-fever, Anaplasmosis, and Erhlichiosis (bacteria) [2,3]. Among the vectors, *Anopheles*, *Aedes*, and *Culex* mosquitoes are of utmost concern as they transmit pathogens that are detrimental to human and animal health, notably malarial and filarial parasites, and a number of arthropod-borne viruses (arboviruses) that cause various encephalopathies in the most critical cases [4,5].

### 1.1. A Glimpse of the Toll of Mosquito-Borne Diseases in Human Populations

The extent of the global burden of mosquito-vectored diseases is remarkably exemplified by only a few diseases that are endemic in the African, Asian, and South American continents. Consider malaria, a protozoal disease of the liver and red blood cells, inflicted by *Plasmodium* species and transmitted by *Anopheles* species. A recent WHO report indicates that in 2022, nearly half the world’s population (>3.9 billion) continued to be at risk of the disease, with an estimated morbidity of 249 million and mortality of 608,000 [6]. Tragically, even with current vector control programs and medical interventions, death rates remain relatively high when compared to other arthropod-borne diseases. Moreover, the daily livelihood is challenging for many who recover from cerebral malaria, which affects >500,000 and kills ~400,000 children annually in Africa, and which is the most severe complication imposed by *Plasmodium falciparum*. Many survivors have an increased risk of neurological, cognitive, and behavioral disorders, and epilepsy. In fact, the leading cause of childhood neurodisabilities in sub-Saharan Africa is cerebral malaria [7,8]. Although advances have been made in integrated pest management strategies, globally, 85 countries and territories remain endemic for malaria, including 45 within the WHO African Region (AR). In 2022, the AR continued to share a disproportionately high level of the global malaria burden, where the morbidity was 233 million (94%) and mortality was 580,000 (95%), and where children under 5 years of age accounted for ~80% of all malaria deaths. Unequivocally, malaria has retained its status as the most significant parasitic disease of humans.

Similarly, arboviruses are also of serious public health concern. These viruses have a wide geographic distribution in tropical and subtropical regions. Currently, over 500 arboviruses have been cataloged, of which ~150 cause, or are implicated in, urgent and neglected tropical diseases (NTDs) [9,10]. These arboviruses belong to the *Flaviviridae*, *Togaviridae*, *Asfarviridae*, *Orthomyxoviridae*, *Reoviridae*, *Rhabdoviridae*, and *Bunyaviridae* families [4,5]. Notable examples of members of the *Flaviviridae* include dengue, yellow fever, and Zika viruses, whereas the Chikungunya virus is a member of the *Togaviridae*. Although Zika virus, arguably an NTD agent, has received much attention in recent years due to its reemergence in Brazil in 2015 and its association with brain abnormalities, including ventriculomegaly, cortical atrophy, calcifications, corpus callosum anomalies, and microcephaly in fetuses [11], dengue virus is by far the most prevalent among arboviruses. Indeed, dengue is endemic in more than 100 countries, mostly in the southern hemisphere [12]. Accurate statistics on the global burden of dengue are uncertain. Nevertheless, the WHO estimates that of the ~3.9 billion people at risk of dengue and dengue fever, ~390 million cases occur annually, of which 96 million are clinically symptomatic [13].

### 1.2. A Glimpse of the Toll of Mosquito-Borne Diseases in Animals

#### 1.2.1. West Nile Virus (WNV)

Whereas it may be natural to focus on mosquito-borne diseases that plague humans, the toll on animals, including commercial farm animals, cannot be underestimated or ignored. The gravity of the problem is illustrated by the unexpected emergence of West Nile virus (WNV) in New York, United States, in 1999. The virus was first identified in Uganda in 1937, is endemic in Africa, the Middle East, and South Asia, occurs sporadically in Europe, and is vectored by bird-feeding mosquitoes, mainly *Culex* species [14]. Enzootic and zoonotic transmissions are responsible for disease in birds and horses, among other animals and humans, in all of which lethal neuroinvasive pathologies could occur [15,16,17]. Shortly after the 1999 introduction and outbreak of WNV on the east coast of the United States, human and animal cases were reported in all states except Hawaii and Alaska, and in Canada [18]. Although mortality in humans remains low, a number of lethal cases have been reported in birds, horses, sheep, reptiles, cats, and rodents [19].

The burden of WNV in animal populations is likely underestimated as reliable morbidity and mortality data collected from mass surveys of the disease in feral animals in deep forests and densely wooded areas are lacking. This presumption is supported by recent surveys demonstrating that among 30 flaviviruses evaluated, WNV has the highest host species diversity, encompassing at least 194 birds and other animals that are likely both targets and reservoirs for the virus [20,21]. Although it is unknown how WNV was introduced to the United States, several hypotheses have been proposed, most implicating animals, including the importation of viremic exotic and migratory birds, infected mosquito eggs, larvae, or pupae, or adult female mosquitoes inadvertently transported from endemic regions, or even as a “trojan horse” in an infected human [22,23].

Finally, and perhaps unfortunately, even though WNV is still responsible for significant morbidity and mortality due to its high rates of transmission in tropical and subtropical regions, Ronca et al. [17] have suggested that it has become an NTD, based on neglect by policy makers and a decline in research and funding.

#### 1.2.2. Japanese Encephalitis Virus (JEV)

Another informative example is the introduction of the Japanese encephalitis virus (JEV) on the Australian mainland. JEV is transmitted principally by *Culex* species, and the disease is endemic in Asia-Pacific tropical and temperate regions, which are home to ~3 billion people. In humans, the annual morbidity is ~100,000 and the mortality is ~25,000, and the disease remains a leading cause of lethal encephalitis in Asia [24]. Pigs are also susceptible to JEV, and as severe infections can result in encephalopathies and reproductive failures, the spread of the virus in piggeries poses a severe economic threat to local farming and commercial industries [25,26]. Moreover, transmission from swine reservoirs facilitated by mosquitoes to dead-end human and horse hosts is of concern.

Before 2021, JEV was largely confined to islands of the Torres Straits and the peak of the Cape York Peninsula. A human case was detected in the Tiwi Islands, and a year later, outbreaks occurred in pig farms in southern Queensland and other regions on the continent, followed by 42 human cases with 7 deaths (16.7%). As a result, a program was initiated to control mosquitoes around piggeries and to vaccinate at-risk human populations, estimated to be ~850,000 based on modeling studies [27]. Due to these events in Australia, the Swine Health Information Center of the US Department of Agriculture Animal and Plant Health Inspection Service (USDA APHIS) initiated steps to explore and mitigate the potential JEV threat in the United States [28].

In summary, these events and scenarios are indicative of challenges that arise with the inadvertent introduction, transmission, and rapid dissemination of potentially lethal pathogens in animals. Perhaps it is likely these events occur at unnoticeable levels in less populated areas or in smaller farms and isolated villages where a limited spread to the local animal and human population is inconspicuous.

## 2. The Expanding Mosquito Range, Climate Change, and Computational Modeling

The few examples noted above clearly demonstrate that the occurrence and proliferation of classic vector-borne tropical diseases in regions where they are normally not problematic cannot be underestimated. Perhaps unsurprisingly, it is now estimated that more than 50% of all known infectious diseases in humans are exacerbated by climate change, in which the increasing temperature expands the lateral and elevated ranges and habitats of a wide variety of arthropod vectors [29,30,31,32,33,34,35].

The reasons for the prevalence of mosquito-borne malaria, dengue, and the reemergence and outbreaks of chikungunya, Zika, and West Nile fevers are complex and compounded by environmental drivers, including climate change and dynamic population flow facilitated by mass migration, travel, and trade from endemic to non-endemic regions [36,37,38,39,40]. Climate change refers to long-term shifts in weather patterns and temperature. It is primarily caused by human activities that generate greenhouse gasses. The environmental accumulation of carbon dioxide, methane, nitrous oxide, and fluorinated gasses has increased exponentially since the preindustrial period as a direct result of human activities [41].

Changes in climatic conditions, especially increases in temperature and humidity, are known to influence the life cycle of vectors and pathogens, directly and indirectly, by altering their fecundity, the pathogen development in the host, transmission rates, and prolonging transmission seasons [39,42]. For example, infection and dissemination rates of West Nile virus in *Culex. p. quinquefasciatus* (Linnaeus) increase at elevated temperatures [43], and the rearing of *Aedes albopictus* (Skuse) at low temperatures (20 °C) decreases these rates by 21% [44].

Computational modeling based on the biological properties of disease-carrying vectors and their changing habitats and known and predictive adaptations to weather patterns, among other factors, have increased markedly over the past two decades. Using a mosquito model system, Couper et al. [45] employed evolutionary rescue theory [46,47] to support the view that the short mosquito generation time, high population growth rates, and strong temperature-imposed selection each favor thermal adaptation. Moreover, when compared to 2021 levels, the model predicted that to maintain a similar level of dengue transmission under the constraint of theoretical 2080 temperatures, the critical thermal maximum for *Aedes aegypti* (Linnaeus) fecundity would need to increase by an average of 1.57 °C (0.03 °C/year).

Evidence supporting the theory that increasing temperature expands the lateral and vertical geographic dispersion of mosquitoes was documented by Carlson et al. [48]. The authors analyzed data on the range limits of *Anopheles* species, vectors that transmit malarial parasites, and the O’nyong-nyong virus that causes fever and polyarthritis [49], collected by medical entomologists over a period of ~120 years (1898–2016). The study represents the largest reliable survey recorded for any formidable vector of human disease. Interestingly, using a regression approach, it appears that these arboviral and parasite vectors gained an average elevation and southward shift of 6.5 m and 4.7 km per year, respectively [48], coinciding with the current 1.2 °C increase in temperature relative to the pre-industrial period [50].

Theoretical models have also been developed to assess the daily abundance of *Aedes aegypti* and *Aedes albopictus*, the Asian tiger mosquito, based on surveillance data collected at various locations in Europe and the Americas between 2007 and 2018 to quantify the propagation and prevalence of dengue, Zika, and chikungunya viruses [51]. The analyses indicated that in regions where both species were present, *Aedes aegypti* was the major vector for transmission of the three viruses, with the “highest risk” occurring in Central America, Venezuela, Colombia, and central-east Brazil, and a “non-negligible risk” of transmission in Florida, Texas, and Arizona in the United States. Significantly, the study suggested that because of its expanding niche, *Aedes albopictus* could contribute to the emergence of chikungunya in temperate regions of the Americas and Mediterranean regions in Europe, primarily Italy, southern France, and Spain.

The range expansion prediction for *Aedes albopictus* in Europe should not be surprising. *Aedes albopictus* was initially confined to Southeast Asia, but as a result of travel and trade due to globalization in recent decades, it is now present in all populated continents. The species apparently was first noted in Albania in 1979 and Italy in 1990, and has since spread to at least 20 countries on the continent. The vector was responsible for dengue and chikungunya outbreaks in Italy, France, and Croatia [52,53].

The extent to which climate change exacerbates this problem remains to be resolved. Nonetheless, in a recent theoretical study by Oliveira et al. [54], using criteria such as current and future climate change projections, population density, and the expectation that the species will invade northern and eastern Europe, there seems to be a consensus that ~83% of urban areas could become suitable habitats for *Aedes albopictus*, compared to ~49% in the current setting. In the future, affected regions could include areas northwest of the Iberian Peninsula, southern France, Italy, and the coastline spanning the western Balkans and Greece. Whereas most cities in Europe were either “suitable with low or high uncertainty”, the study predicted that only a few cities were “unsuitable with low uncertainty”, including Arhus, Copenhagen, Gdansk, Riga, Stavanger, Tartu, and Vilnius, for the invasion and establishment of *Aedes albopitcus*.

Computational analyses have also been used to estimate the geographic distribution of *Aedes aegypti* in Ecuador by 2050, considering factors related to the emission of greenhouse gasses and climate change models [31]. The results suggested that under present climactic conditions, the aquatic larval stage of the vector would not be expected to proliferate at high elevations, including the Andes mountain range and the eastern portion of the Amazon basin. In contrast, when future climate change models were applied to the analyses, the data suggested that elevated mountainous terrain will be permissive for larvae. In this scenario, over 12,000 people currently living in transitional areas will be at risk of pathogens vectored by *Aedes aegypti*.

In summary, both practical and in silico modeling support a strong consensus that increasing temperatures due to climate change will select for vector mosquito strains and perhaps pathogens with thermo-adaptive advantages. As in the past centuries, mosquito-vectored diseases will continue to threaten human and animal health and ecosystems in the wild. As such, these theoretical studies are indispensable and will continue to be of critical value not only in predicting vector-pathogen-human/animal interactions, but also in informing the public and human and animal health agencies in planning and executing coordinated measures to abate the spread of mosquitoes and the pathogens they disseminate. Sadly, considering current trends in climate change and globalization, it is likely that economically and medically marginal communities will continue to be those most impacted by mosquito-borne diseases well into the foreseeable future.

## 3. Integrated Pest Management Programs (IPMPs) for Mosquito Control

Effective mosquito abatement and disease prevention strategies employ integrated approaches, and include at least seven components: (1) mosquito surveillance, in which the use of various types of traps are useful for cataloguing the vector species that are present in a geographic region; (2) physical mosquito control or source reductions focused on eliminating mosquito land and aquatic breeding sites; (3) mosquito larval control measures using chemical or biological control; (4) adult mosquito control (aduticiding) using chemical pesticides that target the stage that transmits viral and parasitic agents of human and animal disease; (5) insect resistance monitoring using cage trials and bioassays—a component that cannot be ignored as mosquitoes are extremely adaptable and can have multiple generations in a single transmission season; (6) public education on measures that can be taken to reduce or eliminate potential mosquito breeding sites, how to avoid mosquito bites, and clinical symptoms associated with vector-borne diseases; and (7) accurate record keeping to establish year-to-year trends and breeding sites, and for regulatory compliance [55,56,57,58]. It should be noted that synthetic repellents, such as DEET, icaridin, and permethrin, and natural oils from cedar, cinnamon, catnip, neem, and citronella, can be applied directly to skin and clothing to prevent mosquitoes from biting and foraging [59,60,61,62,63], but apart from preventing transmission when used appropriately, these personal practices have no measurable impact on the proliferation of mosquitoes in nature.

### 3.1. Synthetic Pesticides

Synthetic pesticides traditionally used for mosquito control include organophosphates (e.g., malathion and Naled) and carbamates, synthetic pyrethroids (e.g., permethrin, resmethrin, and sumithrin), cyclodienes, and organochlorides, including dichlorodiphenyltrichloroethane (DDT), which is banned or has restricted use in many countries [64]. Organophosphates and carbamates inhibit acetylcholinesterase, which leads to a buildup of acetylcholine (ACh) in the synaptic cleft, whereas pyrethroids and DDT are neurotoxins that preferentially target voltage-gated sodium channels, leading to excitatory paralysis [65,66,67]. Cyclodiene insecticides, such as dieldrin, and phenyl pyrazoles, such as fipronil, target GABA (γ-aminobutyric acid) receptors, subsequently leading to a decrease in the stimulation of neurons [68].

Apart from the more traditional pesticides, others used in recent years include neonicotinoids, spinosyns, pyrroles, and insect growth regulators (IGRs). Neonicotinoids, such as nicotine, bind to nicotinic ACh receptors (nAChRs) and include dinotefuran and clothianidin, which show promise in mosquito control [69,70]. Under normal physiological circumstances, a low-to-moderate activation of nAChRs by ACh elicits nervous stimulation, whereas high levels of the neurotransmitter overstimulate and block these receptors, resulting in paralysis and death. Unlike Ach, which is broken down by acetylcholinesterase, the enzyme has no effect on neonicotinoids. The pesticide binds irreversibly to the enzyme, leading to the paralytic death of the insect [71]. Spinosyns are metabolites produced by the soil bacterium, *Saccharopolyspora spinosa*. Members of the spinosyn family of insecticides, including Spinosad, which is composed of spinosyns A and D, have a unique mode action in that they disrupt AChR in a wide variety of arthropods, including mosquitoes, particularly *Aedes* and *Culex* [72,73,74,75].

Pyrroles and IGRs are viable alternatives to neurotoxins in mosquito control. Pyrroles, including chlorofenapyr, are broad spectrum insecticides, which, unlike neurotoxins, disrupt respiratory pathways and proton gradients through the uncoupling of oxidative phosphorylation in the mitochondria, and are effective in bed nets and indoor treatments for the control of *Anopheles*, *Culex*, and pyrethroid-resistant *Aedes aegypti* [76,77,78,79]. The use of IGRs in IPMPs are attractive because of their low toxicity to mammals and non-target species. IGRs elaborate their effect by disrupting insect development; for example, methoprene mimics juvenile hormones and prevents larvae from completing their immature stage, thereby reducing the adult population, and pyriproxyfen inhibits chitin synthesis, which is essential for formation of the exoskeleton of insects [80,81,82].

Synthetic pesticides are generally quite effective mosquito adulticides. Despite the rapid kill they induce, unintended negative environmental and ecological impacts cannot be ignored. These chemicals can directly harm non-target invertebrate and vertebrate species, accumulate in the environment, affect food webs for protracted periods, and, in particular, impose selective pressures leading to the persistence of resistant mosquito populations [83,84,85]. Regarding the latter, resistance to organophosphates, including larvicidal temephos and chlorpyrifos, carbamates, organochlorines, and pyrethroids, has been documented for species of *Aedes*, *Culex*, and *Anopheles* [86,87,88,89,90]. Taken together and apart from climate change and globalization, resistance to synthetic insecticides is a major contributor to the proliferation of mosquitoes and spread of infectious diseases globally.

In an excellent review by Liu [83], two major mechanisms for insecticide resistance were addressed, i.e., target-site insensitivity and increased metabolite detoxification. Regarding the latter, it is well-established that insect cytochrome P450 monooxygenases, esterases, and glutathione S-transferases (GSTs) play significant roles in the detoxification of plant toxins and xenobiotic, such as natural and synthetic insecticides. Generally, (i) monooxygenases are involved in the metabolism of virtually all insecticides; (ii) esterases can metabolize organophosphates, carbamates, and pyrethroids, which are rich in ester bonds; and (iii) GSTs can neutralize the effect of pyrethroids, organochlorides, and organophosphates. Currently, factors leading to the hyperexpression of genes coding for monooxygenases, esterases, and GSTs in response to synthetics in mosquitoes are of considerable interest [91,92,93,94,95].

Regarding the insecticide target site specificity, certain site-specific mutations in sodium channel proteins, acetylcholinesterases, and GABA receptors are strongly associated with resistance to their ligand pesticides. For example, studies by Xu et al. [96] and Li et al. [97] on the sodium channel of *Culex quinquefasciatus* showed that at least three specific nonsynonymous mutations (A109S, L982F, W1573R) were directly associated with resistance to permethrin, and that six synonymous mutations (codons for L582, G891, A241, P1249, and G1733) that do not alter the amino acid sequence may play a role in the evolution of resistance. In *Culex* and *Anopheles*, in addition to other insect species that display an insensitivity or a reduced sensitivity to organophosphates and carbamates, a mutation in the *ache1* gene conferring a G119S substitution likely causes steric hindrance that reduces the accessibility of the inhibitor pesticide substrate to acetylcholine esterase 1 (AChE1) [83,98,99,100,101]. The major neuronal inhibitory mechanism in insects (and vertebrates) is the GABAergic system, in which activation suppresses neuronal excitability. The GABA receptor is targeted by dieldrin and fipronil, which are cyclodiene and phenyl pyrazole insecticides, respectively. Mutations resulting in A296S/G substitutions in the GABA receptor are associated with dieldrin resistance in many insects, including *Anopheles gambiae* (A296G), *Anopheles arabiensis*, *Anopheles stephensi*, *Anopheles funestus*, and *Aedes aegypti* (A296S), and generally lower levels of resistance to fipronil [83,102,103,104,105,106,107].

In other regards, it is interesting to note that the GABAergic system also plays an important role in immune regulation in mammals, for example, in the autoimmune inflammation and migration of immune cells in response to parasitic infection with *Toxoplasma gondii* [108,109,110]. It is now apparent that a similar role for GABA signaling occurs in mosquitoes. Zhu et al. [111] showed that (i) the dsRNA-mediated disruption of GABA and the specific inhibition of GABA_A_ receptor decrease arboviral replication, whereas the introduction of glutamic acid per os increases the ability of arboviruses to infect mosquitoes; (ii) blood meals enhance viral replication through GABAergic activation; and (iii) the GABAergic system suppresses the Imd pathway, an NF-kB pathway known to regulate bacterial and malarial infection in mosquitoes [112,113,114]. Given this scenario, the extent to which sublethal levels of insecticides dampen or inhibit the GABAergic system, and how resistance to these synthetics influence the propagation of pathogens in natural mosquito populations, remain to be resolved. Nevertheless, Zhu et al. [111] demonstrated that at least two GABA inhibitors, fipronil and bilobalide, markedly reduced dengue (DENV-2) and Zika virus loads in *Aedes aegypti* that survived treatment with these chemicals, a finding that suggests that inhibitors of the GABAergic system may play a role in reducing the dissemination of arboviruses in the field.

### 3.2. Avoidance of Pesticide through Olfaction

Whereas most studies on mosquitoes focus on the diseases they transmit and control strategies, comparatively less attention has been paid to olfaction in this group of dipterans [115], specifically as it relates to ‘learned avoidance’ of pesticides using the World Health Organization (WHO) standard tube bioassay [116,117]. Toward this end, a recent study by Sougoufara et al. [118] demonstrated that female *Aedes aegypti* and *Culex quinquefasciastus* exposed to sublethal doses of five synthetic pesticides used in vector control, i.e., deltamethrin (pyrethroid), permethrin (pyrethroid), lambda-cyhalothrin (pyrethroid), propoxur (carbamate), and malathion (organophosphate), exhibited associative learning behavior. Female mosquitoes previously exposed to a chemical avoided the same chemical when associated with adverse survival odds while seeking out blood meals to ensure survival. The study highlights the possibility that under natural conditions, following a single exposure, mosquitoes can associate the smell of pesticides with their harmful effect and avoid contact with the said chemical. The ability, or the potential of mosquitoes to evade pesticides in the field through associative learning mediated by olfaction is perhaps one explanation for the seasonal proliferation of these vectors and underscores the necessity for the compensating measures in IPMPs.

## 4. Effective “Green” Technologies

It is well-established that natural predators, such as copepods, water bugs, fish, and amphibians (e.g., frog tadpoles), that feed on larval and pupal aquatic stages of mosquitoes play a role in the ecology of these arthropod vectors. Larvivorous fish, including *Gambusia* and *Poecilia*, have been used in over 60 countries. In spite of their potential threat to native aquatic fauna, these predatory fish have been effective in decreasing larval and pupal populations, thereby lowering the adult mosquito burden and, by extension, disease transmission in the regions where they have been employed [119,120].

More targeted eco-friendly mosquito control strategies include the classical Sterile Insect Technique (SIT), which utilizes a large-scale release of irradiated adults; the Incompatible Insect Technique (IIT), in which artificially increasing *Wolbachia* infection levels in adults impose fitness costs; and the Release of Insect carrying a Dominant Lethal gene technique (RIDL), which selects against daughter progeny [121,122,123,124]. A CRISPR/Cas9 gene drive system has also been proposed for pest and vector insect population control [125]. In addition, naturally occurring bacterial larvicides that destroy larval midgut have proven to be successful as biocontrol agents. The two most notable bacteria are *Lysinibacillus sphaericus* (Ls, formerly *Bacillus sphaericus*) and *Bacillus thuringiensis* subsp. *israelensis* (Bti). The larvicidal proteins are the binary toxin Tpp1Aa1/Tpp2Aa1 (formerly BinA/BinB) of Ls, and Cry4Aa1, Cry4Ba1, Cry11Aa1, and lipophilic Cyt1Aa1 [126] of Bti packaged in a unique composite prokaryote insect larvicidal organelle (PILO) [127]. Each of these control strategies is intended to ultimately prevent the production of viable offspring, thereby suppressing mosquito populations and the subsequent transmission and dissemination of viral and parasitic agents of disease.

### 4.1. Sterile Insect Technique (SIT)

The application of the SIT has its origin in the 1950s when it was used to control insect pests in agriculture [128,129]. For mosquito control, specific vector species are reared and mated in insectaries. The resultant male and female pupae are separated, and, subsequently, males that emerge are exposed to ionizing X-ray or gamma radiation that induces deleterious dominant mutations in germ cells. Sterile males are released on a mass scale to compete with wild competent males in the field. Following mating with sterile males, females lay sterile eggs that do not develop into progeny, leading to a reduction in species-specific vector populations and the desired effect of lowering disease burdens on human populations. The SIT has been used since the 1960s and more recently with measurable success to control *Anopheles*, *Aedes*, and *Culex* species in the United States, Asia, Central America, and Cuba, and plans are underway for its application on the African continent [130,131,132,133,134,135]. Nevertheless, the widespread use of the SIT is hampered by a number of formidable factors, including mass production, sex separation, and the continuous release of sterilized males for effective suppression of robust populations in the field [122,136].

### 4.2. Wolbachia and Incompatible Insect Technique (IIT)

*Wolbachia pipiens* is an endosymbiotic bacterium that occurs in the cytoplasm of approximately 60% of insects and is maternally (vertically) transmitted to offspring [137]. Bacteriophages found in *Wolbachia* are primarily responsible for the phenomenon of cytoplasmic incompatibility (CI), the most common type of reproductive interference in insects. *Wolbachia* phages harbor incompatibility determinants, including *cifA* and *cifB*, that regulate CI such that when male insects that have both *cifA* and *cifB*, and, in some systems, just *cifB*, mate with females lacking *cifA*, no viable offsprings are produced [138,139]. The IIT approach takes advantage of the fact that male mosquitoes infected with the natural endosymbiont are unable to produce offspring with female mosquitoes devoid of the bacterium or that have cytoplasmic incompatible strains of the bacterium [124,140,141]. Although females can lay eggs, the eggs never hatch. Over the past decade, *Wolbachia*-based intervention programs have focused primarily on reducing *Aedes* species transmitting dengue virus in affected regions in India, Malaysia, China, Singapore, Australia, Brazil, and in the United States where, since 2017, the CI-based MosquitoMate ZAP pesticide has been permitted for use in at least 20 states [142]. Where data are available for the period surveyed, reductions in the incidence of dengue ranged from 40% in Kuala Lumpur, Malaysia, to 96% in Cairns, Australia [143,144,145,146,147,148,149,150,151,152,153].

Although *Wolbachia* is known to be an endosymbiont in *Culex* and *Anopheles* species, studies have not been conducted to determine the efficacy of the IIT in reducing natural populations of these mosquitoes or the impact on the incidence of viral and parasitic agents of diseases they transmit on a large geographical scale. It is known that suppression is mediated by *Wolbachia* incompatibility in *Culex pipiens fatigans*, and there appears to be promise for controlling *Culex quinquefasciatus* in the field [154,155,156,157]. The use of *Wolbachia* in mosquito control is also attractive because the intracellular bacterium can be transmitted vertically and spread horizontally among field populations. As several studies suggest that *Wolbachia* can inhibit the proliferation of *Plasmodium falciparum* in *Anopheles stephensi* and *Anopheles gambiae*, and West Nile virus in *Culex quinquefasciatus* [158,159,160], targeted control utilizing the appropriate strains of *Wolbachia* could be helpful in suppressing pathogen propagation in these mosquitoes.

#### *Wolbachia* Can Enhance Pathogen Proliferation

On the other hand, while the utility of *Wolbachia* in controlling mosquito populations has been successful, there are concerns that may limit its fidelity and efficacy. Several studies have shown that *Wolbachia* can enhance rather than suppress pathogen proliferation in insects, including the *Plasmodium* parasite and West Nile virus, and even insect-specific flaviviruses that are not etiologic agents of human disease [161,162,163,164,165,166,167,168,169]. Considering climate change, globalization, and the geographical expansion of mosquitoes, it is reasonable that these apparently conflicting issues should be addressed adequately by a diversity of experts, including ecologists, before large-scale *Wolbachia*-based control programs become routine.

### 4.3. Combined SIT–IIT

The objective of the IIT can be undermined by the inadvertent, accidental release of female mosquitoes that harbor the same *Wolbachia* strain present in the released male counterpart. Random mating with compatible sexes will result in viable offspring in the field. To address this issue, Zheng et al. [146] showed that by combining the SIT and the IIT, an almost complete elimination of *Aedes albopictus* in two isolated islands in Guangzhou, China, occurred over a two-year period. Although certain objections have been raised regarding this study [170], data from other field experiments have provided credence to the approach, including the suppression of *Aedes aegypti* in semi-rural Thailand [171]. Use of the combined technique has been proposed elsewhere, including Singapore, Hawaii, and Mexico [172,173,174].

### 4.4. Release of Insect Carrying a Dominant Lethal Gene (RIDL)

RIDL utilizes engineering techniques for the production and release of genetically modified mosquitoes that harbor genes that are lethal to their young offspring. The lethal gene is regulated by a molecular switch that is turned off during the mass production of the RIDL insects in insectaries. The switch utilizes a tetracycline-repressible expression system in which the tetracycline-repressible transcriptional activator protein (tTA) is placed under the regulation of a selected promoter that governs essential developmental specificities of the insect. The development-specific expression of tTA results in the tTA activator binding to a specific sequence, *tetO*, driving the expression of a lethal effector protein gene from a minimal promoter, and subsequently leading to the death of the host in which it is expressed. The system is easily manipulated when insects are reared in the presence of low levels of tetracycline, which disrupts the binding of tTA to *tetO*. Without tetracycline, RIDL adult males develop normally, and females die in the preimaginal stage of development. The selected lethal gene is turned on in the absence of tetracycline when the engineered males are released [175,176,177,178,179]. The RIDL adults then mate with their wild counterparts, and the resulting female offspring die in the larval and pupal stages when the lethal gene is expressed. Repeated mating in the field leads to a decline in females and, subsequently, a potential population crash. RIDL has been used successfully to control lepidopteran pests of agriculture and vector mosquitoes in cage studies and in the field [122,178,180,181,182,183,184,185,186,187,188,189,190,191].

#### RIDL OX513A—Success and Concerns

As alluded to above, the use of “self-limiting genetic technology” to control vector mosquitoes, where used, has already proven to be successful. This is more specifically exemplified by the results of sustained releases of RIDL OX513A (Oxitech Ltd., Abingdon, UK). OX513A is a robust commercial fluorescent-tagged transgenic *Aedes aegypti* that harbors a conditional lethal gene engineered to deliver a dominant non-sex-specific deadly effect on targeted natural populations of the species [186]. The release of the OX513A strain in 2010 resulted in an 80% suppression of native *Aedes aegypti* in the Cayman Islands [187,188]. Later, in 2012, the release of the strain in a suburb of Juazeiro, Bahia, Brazil, led to an 81% to 95% suppression of local *Aedes aegypti* within the span of a year [189].

Despite these observations, there are reasonable concerns that lethality may not be complete, and genes can indeed be transferred from engineered strains to native populations that can then spread and be regionally established in successive generations. A study showing this to be true was conducted in Jacobina, Bahia, Brazil [190]. Approximately 450,000 OX513A males were released each week over a 27-month period. Samples were collected 6, 12, and 27–30 months after releases began, and the genotypes of 57 fluorescent larvae collected six months after the initial release that represented hybrid F1 were determined. As the OX513A and native Jacobina population were genotyped for >21,000 single nucleotide polymorphisms (SNPs), accurate genotype SNP assessments convincingly showed that viable hybrids are capable of reproducing in nature, at least within regional confines.

Interestingly, OX513A was developed using a strain that originated in Cuba and outcrossed to a Mexican population. Therefore, extant *Aedes aegypti* in Jacobina represents a mix of the three populations. It must be noted that transgene from OX513A was not found in the hybrids, and there is no evidence to suggest that the hybrids were more robust [142,192]. Exactly how this affects the population ecology and acquisition and dissemination of vectored pathogens in the region remains to be determined. Nonetheless, these potential long-term effects cannot be ignored, especially when taking into account that, depending on the sampling and analytic criterion used to define unambiguous introgression, from 10% to 60% of the population harbor genetic sequences originating from OX513A [190]. (For transparency, the reader is directed to the online rebuttal by Oxitec to the findings of Evans et al. [193].)

### 4.5. CRISPR/Cas9 Gene Drive

Another emerging technology that could prove to be indispensable in mosquito control is CRISPR/Cas9 gene drive. The CRISPR/Cas9 system has received considerable attention over the past decade not only because of its significant role in bacterial and archaeal immunity, but also because of its broad application as a highly specific gene-editing tool [194]. This system utilizes a noncoding guide RNA (gRNA) that allows the Cas9 endonuclease to cleave dsDNA at a designated site. The cleavage is repaired in vivo by non-homologous end-joining or homology-directed repair [195]. From an applied perspective, the CRISPR/Cas9 system can be used to generate mosquitoes that harbor heterologous sequences or deleterious mutations that affect the filial generation in which the targeted function is expressed, or even to potentially disrupt the pathogen lifecycle in biological vectors. For example, sex determination in *Aedes aegypti* is regulated by the M factor, a dominant male-determining factor harbored in the M locus of the Y chromosome. Hall et al. [196] showed that an M-locus gene, Nix, functions as the M factor in *Aedes aegypti*. CRISPR/Cas9 knockout of Nix resulted in genetic females with almost complete male genitalia, demonstrating that the technique could be used to convert female mosquitoes into essentially harmless phenotypic males that are incapable of breeding.

The disruption of functions required for female mosquito development will likely be the hallmark of the gene drive technology. This is highlighted in studies where three genes (AGAP005958, AGAP011377, AGAP007280) that confer female-sterility phenotypes were disrupted with CRISPR/Cas9 constructs; the transmission to progeny rates ranged from 91.4–99.6% [197]. Genetic systems that distort sex ratios with a bias for males via RNA-guided shredding of the X-chromosome during spermatogenesis are also quite promising [198,199,200]. More current cage studies have been described, and the results lend support to the efficacy of this emerging technology in controlling vector mosquito populations [201,202,203,204,205]. In addition, commentaries on the prospect of eradicating malaria using CRISPR/Cas9 gene drive [206,207] and regulatory and policy considerations [208] are the subjects of scrutiny and debate.

### 4.6. CRISPR-Based Engineering of Mosquitoes Refractive to Pathogens

#### 4.6.1. CRISPR/Cas9

Engineering mosquitoes to be refractive to pathogens by disrupting both their life cycles is also promising. Dong et al. [209] showed that disruption of the fibronectin-related protein 1 (FREP1) gene in *Anopheles gambiae* conferred a profound effect in suppressing infection by *Plasmodium falciparum* and *Plasmodium berghei*, which are malarial parasites of human and rodents, respectively. The engineered mosquitoes were less robust in blood-feeding, fecundity, and egg-hatching. Moreover, the fitness cost extended to poor larval and pupal development and a shorter life span after a blood meal.

More recent studies show that disruption of the *Anopheles gambiae* γ-interferon-inducible thiol reductase (mosGILT) gene impaired the ovarian development and the production and accumulation of yolk in the developing oocyst (vitellogenesis), and was less permissive for the human and rodent *Plasmodium* parasites [210]. Although these mutant mosquitoes are partially refractive to malarial parasites, the imposed fitness cost likely excludes further development for their use in malaria control. These mutants will not be able to compete successfully in the wild. In any case, at present, such strategies may not be a practical priority, given the recent successes of SIT, IIT, and RIDL.

#### 4.6.2. CRISPR/Cas13

In 2015, the Class 2 candidate 2 (C2c2) of Cas proteins, which later became known as Cas13a, was identified in prokaryotes. Interestingly, unlike the well-known Cas9, Cas13 proteins are unique in that they specifically bind and cleave ssRNA using a guide crRNA of approximately 64–66 nucleotides. Notably, once activated, Cas13 becomes promiscuous and degrades RNA indiscriminately, and, therefore, can be lethal to the cell in which it is expressed [211,212,213,214]. This feature makes Cas13 attractive from an applied perspective; the system could potentially be used to “kill two birds with one stone”, i.e., suppress the replication of pathogens in the mosquito vector as an initial response to infection and, subsequently, target both host and pathogen RNAs indiscriminately. As a result, the system could be lethal to both the pathogen and the vector, or at least reduce the fitness of the vector. In principle, the transmission and dissemination of pathogens and vector population decline are theoretically possible.

The ‘proof of concept’ application of CRISPR/Cas13 has been demonstrated for Huntington’s disease, an inherited disease that causes the degeneration of neurons in the brain [215], and in *Drosophila* [216] and mosquitoes [217,218]. More recently, Dalla Benetta et al. [219] demonstrated the practical feasibility of CRISPR/Cas13 in *Aedes aegypti*. The research group developed an antiviral strategy called REAPER (vRNA Expression Activates Poisonous Effector Ribonuclease) that uses a sensor switch to activate the system in vivo. The system design allowed REAPER to remain dormant until the engineered mosquito acquired a blood meal. Soon thereafter, the activated expression of four gRNA led to the suppression of chikungunya virus replication, and the collateral effect, though not absolute, in which at least 35% of mosquitoes died post-viral infection.

## 5. Eco-Friendly *Lysinibacillus sphaericus* and *Bacillus thuringiensis* subsp. *israelensis*

The two well-known bacteria used in mosquito control programs worldwide are *Lysinibacillius sphaericus* (Ls) and *Bacillus thuringiensis* subsp. *israelensis* (Bti). These Gram-positive bacteria are spore-forming aerobic to facultative anaerobic bacilli that occur naturally in soil. Although they produce several different proteinaceous and non-proteinaceous toxins during vegetative growth (e.g., Mtx, Vip), they are best known for the proteinaceous parasporal crystalline inclusions (e.g., Cry, Cyt, Tpp) they synthesize during the sporulation phase of growth [220]. The parasporal inclusions of Ls and Bti are composed of different types of protoxins, but, nevertheless, they share a common feature in that when ingested by mosquito larvae, they solubilize in the alkaline environment of the midgut where they elicit substantial damage to the epithelial lining leading to larval death [221].

### 5.1. Brief History of Lysinibacillus (Formerly Bacillus) sphaericus (Ls)

The first mosquitocidal strain of Ls (Neide) was isolated from moribund ‘cool weather mosquito’ *Culiseta incidens* (Thomson) larvae in Fresno, California [222]. Several other strains with varying levels of toxicities against *Culex* and *Anopheles* larvae were subsequently isolated [223,224]. In particular, Weiser [225] isolated a highly toxic strain, Ls (Bs) 2362, in Nigeria that killed *Culex* and *Anopheles* larvae and showed that the lethality was due to the binary toxin, BinA/BinB, which was recently reclassified as Tpp1Aa1/Tpp2Aa1 [126,220]. The Ls 2362 was commercialized in 2000 under the name VectoLex and is also currently used in combination with *Bacillus thuringiensis* subsp. *israelensis* (Bti) in VectoMax (Valent Biosciences, Libertyville, IL, USA)

#### 5.1.1. Structural Characteristics of Tpp1Aa1/Tpp2Aa1

Different Tpps have been isolated from Ls, but the three-dimensional (3D) structure of Tpp1Aa1/Tpp2Aa1 nanocrystals was resolved at 2.25 angstroms de novo using serial femtosecond crystallography (SFX) at an X-ray free-electron laser [226]. Although their amino acid sequence identity (28%) and similarly (46%) have diverged significantly, Tpp1Aa1 and Tpp2Aa1 possess only a few differences at the structural level and have a size of 100 angstroms long and 25–30 angstroms in diameter. Each protein is composed of two domains, a β-trefoil domain, which is involved in carbohydrate- and receptor-binding, and the pore-forming domain, located at the amino- and carboxy-terminals, respectively.

The main structural difference between Tpp1Aa1 and Tpp2Aa1 occurs at the β-trefoil domain, which could be implicated in distinct roles of these proteins during the intoxication process in the midgut epithelia, where Tpp2Aa1 has a less prominent role in carbohydrate binding when compared to Tpp1Aa1. The few structural differences between Tpp1Aa1 and Tpp2Aa1 indicate they can form a heteromeric pore assembly complex with a topology similar to the aerolysin family of pore-forming toxins [227,228]. The crystal structure of other Tpps, such as Tpp49Aa1, Tpp1A2, Tpp80Aa1, and Tpp35Ab1, have been elucidated. Although they have a few differences, they maintain similar structures, i.e., the receptor and pore-forming domains [228].

It must be noted that Ls’s Tpp1Aa1/Tpp2Aa1 is a single-receptor-specific toxin, the receptor being a GPI-anchored amylomaltase in the midgut epithelial membrane of *Culex* and *Anopheles* species Cqm1/Cpm1 and Agm3, respectively, to which advantageous mutations in the host lead to rapid resistance in field populations [223,229,230,231]. Additionally, although a few *Aedes* species are susceptible to Tpp1Aa1/Tpp2Aa1, *Aedes aegypti* larvae are naturally refractive to Tpp1Aa1/Tpp2Aa1 as it lacks a suitable midgut receptor [232]. Therefore, Ls’s use in IPMPs is inconsequential to this species.

#### 5.1.2. Mechanism of Toxicity of Tpp1A1/Tpp2Aa1

The mechanism of toxicity of Tpp1Aa1/Tpp2Aa1 is unclear. Studies show that after exposure to the binary toxin, (i) *Culex quinquefasciatus* larvae stop feeding within 4 h, and body paralysis occurs at 36 h, most likely due to neural and muscular tissue damage; (ii) the binding of Tppp2A1 to the Cqm1 receptor is essential for internalization of Tpp1Aa1/Tpp2Aa1, likely mediated by an endocytic pathway; and (iii) a number of cytotoxic effects follow, including an increase in the lysosomal number and size, damage to the mitochondria, intense cytoplasmic vacuolization, destruction of the endoplasmic reticulum, and apoptosis, collectively leading to the destruction of the midgut microvilli [233,234,235,236,237,238]. Transcriptome analyses of *Culex quinquefasciatus* following intoxication with the binary toxin show a downregulation of genes related to metabolism and mitochondrial function, and an induction of genes coding for proteins linked to mitochondrial-mediated apoptosis, autophagy, and lysosomal compartments [239]. The collective data suggest that diverse pathways are involved in cytotoxic and tissue tropic (neuromuscular) malfunctions that culminate in larval death.

### 5.2. Brief Historical Account of Bacillus thuringiensis Strains—A Collective of Highly Specific Insect Larvicides

The first isolate of *Bacillus thuringiensis* was discovered over 120 years ago in Japan by Ishiwata Shigetane, a sericultural engineer, in larvae of the silkworm moth (*Bombyx mori*), in which it caused a sudden lethal disease, bacillary paralysis [240]. Fourteen years later, Berliner [241] described a similar disease in larvae of the flour moth, *Ephestia kuhniella*, in Thuringia, Germany, hence the species name “*thuringiensis*”. By the mid-1970s, based on the biological profiles of hundreds of isolates, at least thirteen subspecies were characterized. These Bt subspecies exhibited larvicidal activities against a broad range of lepidopterous (moth) and coleopterous (beetle) larvae, and a few were developed commercially as formidable eco-friendly biopesticides. These include Bt subsp. *kurstaki* (Btk; Garden Dust, Caterpillar Killer, Dipel); *Bt* subsp *aizawai* (Bta; XenTari BT DF, Certan B 401), which targets lepidopteran pests; and Bt subsp. *morrisoni* strain *tenebrionis* (Btm), which is toxic to coleopteran pests [242].

Like Ls, *Bacillus thuringiensis* is an aerobic Gram-positive spore-forming rod that is naturally present in many ecological niches, including soil, plants, stored products, aquatic environments, and insects and their habitats [243]. The bacterium is classified in the *Bacillus cereus sensu lato* (*sl*), which contains at least 22 non-pathogenic and pathogenic species, with the most notable of the latter being *B. anthracis* [244]. Although these bacteria share a high degree of genetic identity, Bt is distinguished from other members of *B. cereus ls* by the parasporal crystalline inclusions. These inclusions are composed of larvicidal protein protoxins that are produced during the sporulation phase of growth. In addition to the established subspecies and strains, a growing list of new Bt isolates has led to the identification of novel protein toxins, which complicates their classification. Nonetheless, Crickmore et al. [126,220] have developed an informative conserved structure-based nomenclature system that includes over 1100 Bt crystalliferous and other bacteria-derived pesticidal proteins produced during sporulation and vegetative growth, including VIPs (vegetative insecticidal proteins) and Mtx (mosquitocidal toxins) initially isolated from *Lysinibacillus sphaericus*. Fifteen classes (Cry, Cyt, Vip, Tpp, Mpp, Gpp, App, Spp, Mcf, Mtx, Vpa, Vpb, Pra, Prb, and Mpf) have been defined based on conserved domain similarities, and a separate class (Xpp) has been set aside for pesticidal proteins with unknown or uncharacterized structures.

The Cry (crystalline; ~70–140 kDa) and Cyt (cytolytic, ~24–27 kDa) proteins are the most studied and are the basis for the most successful commercial Bt-larvicide products used in the biocontrol of lepidopterans, coleopterans, and dipteran (mosquito and blackflies) pests, Btk HD1 (Cry1Aa, Cry1Ab, Cry1Ac and Cry2Aa) and Bta (Cry1Aa, Cry1Ab, Cry1Ca and Cry1Da), Btm (Cry3Aa and Cry3Ba), and Bti (Cry4Aa, Cry4Ba, Cry11Aa and Cyt1Aa), respectively [221]. Moreover, it must be noted that agro-industries have exploited and re-engineered many *cry* and *vip* genes for expression in transgenic crops. Excellent recent reviews have been published on genetically modified crops, including Kumar et al. [245], Yamamoto [246], and Gassmann and Resig [247].

#### 5.2.1. *Bacillus thuringiensis* subsp. *israelensis*, Bti

Despite the trove of Bt isolates cataloged by the mid-1970s, none were active against dipteran larvae. This changed in 1976 with the isolation of Bt (60A) in Israel. The bacterium was isolated from dead *Culex pipiens* larvae found in a stagnant pond in the north central Negev Desert and it was shown to be lethal to larvae of Nematoceran insects (e.g., mosquitoes, blackflies, and chironomid midges) [248]. The *Bt* subsp. *israelensis* (Bti) 60A isolate initially demonstrated rapid toxicity against larvae of five different species assayed, i.e., *Anopheles sergentii* (Theobald), *Uranotaenia unguiculata* Edwards, *Culex univitattus* Theobald, *Aedes aegypti* and *Culex pipiens*, and the activity was 30–100 times greater than that of *Lysinibacillus* (*previously Bacillus*) *sphaericus* SSI-1.

Shortly after the safety and efficacy of Bti were demonstrated, commercial products were developed for applied use worldwide [249]. At present, at least 26 different formulations (wettable powders and suspensions, granules, and briquettes) based on Bti, or a combination of Bti and Ls, are used in mosquito control programs globally, including, Vectobac, Bactimos, ABG6138G, and Teknar; and VectoMax, Culinexcombo, FourStar, and BTBSWAX, respectively [250,251].

#### 5.2.2. Bti, the Most Robust and Efficacious Natural Mosquito Larvicidal Bacterium Known

To date, the Bti ONR 60A serotype H-14 strain is the most widely used and environmentally safe bacterial larvicide, primarily targeting *Aedes*, *Anopheles*, and *Culex* species [252]. It is also used to control *Simulium* species (blackflies) that vector *Onchocerca volvulus*, the etiological agent of river blindness (onchocerciasis) [221]. River blindness is endemic in Africa, and the disease is also known to occur in at least six countries in the Americas (Brazil, Colombia, Ecuador, Venezuela, Guatemala, and Mexico), where the parasite was introduced as a result of the slave trade [253]. Although significantly less prevalent than mosquito-borne viral and parasitic diseases, onchocerciasis is listed as an NTD. Recent estimates of onchocerciasis are still alarming. At least 390,000,000 people in 31 countries required preventative treatment with ivermectin, 14.6 million of those infected presented with skin diseases, and 1,150,000 had vision impairments [3,254].

Interestingly, it must be noted that the expanded host range for Bti includes other dipterans, such as the Mexican and Mediterranean fruit flies and fungus gnats [255,256], and pea and potato aphids, which are hemipteran and homopteran pests, respectively [257,258,259,260], and coleopteran cotton boll weevil and leaf beetle [261,262]. Interestingly, non-arthropod targets of Bti are the cercariae stages of human and avian parasitic flukes *Schistosoma mansoni* and *Trichobilharza szidata*, respectively, which are susceptible to the water soluble M-exotoxin [263], the intermediate *Oncomelania* snail host of *Schistosoma japonicum*, when assayed at unusually high concentrations (900 ng/mL) [264], and the root-knot nematode, *Meloidogyne incognata* [265].

#### 5.2.3. Structural Characteristics of Bti’s Larvicidal Proteins

As mentioned above, the main larvicidal components of the Bti’s prokaryotic insect larvicidal organelle (PILO) are crystalline inclusions of Cry4Aa1, Cry4Ba1, Cry11Aa1, and Cyt1Aa1 (Figure 1) [127].

The genes coding for these proteins are harbored on plasmid pBtoxis [267]. The crystallographic structures of the four toxins have been determined. Cry4Aa1, Cry4Ba1, and Cry11Aa1 contain the typical three-domain structure of Cry toxins. Domain I, located in the N-terminal, is an amphipathic α-helical bundle responsible for oligomerization, membrane insertion, and pore formation, whereas domain II is formed by antiparallel β-strands harboring loops that participate in receptor binding and specificity. Domain III is a sandwich of two antiparallel β-sheets involved in receptor binding and protection of the toxin’s structural integrity. Although the three domains among Bt larvicidal toxins are very similar, they possess structural differences, but in comparison to domains I and III, domain II is the most divergent, which supports its role in toxin specificity. It is interesting that Cry4Ba1, a diptera-specific protein, is more closely related to Cry1A (lepidopteran-specific) than Cry3Aa (coleoptera-specific), but is less closely related to Cry2Aa (lepidoptera/diptera specificity) [268,269,270].

The 3D structure of Cyt1Aa1 was elucidated at 2.2 A resolution. The toxin has a cytolysis fold with central β-sheets surrounded by two outer α-helical layers that can undergo conformational changes. The α-helical layers swing away to allow the exposed β-sheets to insert into the membrane. Lipid-binding moieties have been identified between the β-sheets and the α-helical layers and highlight their affinity for membrane lipids once activated in the larval alkaline midgut [271].

#### 5.2.4. Mechanism of Toxicity of Bti’s Larvicidal Proteins

Numerous studies and reviews have been published on the mechanisms involved in the toxicity of larvicidal Cry proteins [272,273,274,275,276]. In summary, as with Ls binary toxin, the protoxins that compose the crystalline inclusions of Bti’s PILO are solubilized and proteolytically activated in the alkaline midgut of mosquito larvae. The activated Cry toxins disrupt the membrane integrity and osmotic balance by creating pores through interactions with midgut microvillar membrane receptors and binding to plasma membrane adhesion proteins. Several receptors have been identified and include cadherins, aminopeptidase N, alkaline phosphatases, and α-amylase.

Unlike Bti’s Cry4Aa1, Cry4Ba1, and Cry11Aa1, which elicit cytotoxicity through a receptor-mediated mechanism, specific membrane-associated receptors have not been identified for Cyt1Aa1. Nonetheless, activated Cyt1Aa1 is intrinsically highly lipophilic and preferentially binds unsaturated fatty acids. To prevent the destruction of the plasma membrane in vivo, the synthesis of Cyt1Aa1 requires a 20-kDa helper “chaperone” protein that is also encoded by pBtoxis, where the corresponding gene is a component of the *cry11Aa1* operon [277,278,279]. It is thought that binding of activated Cyt1Aa1 to lipids directly perturbs and destabilizes the plasma membrane integrity [271,279,280,281].

Interestingly, Cyt1Aa1 is considerably less toxic than Cry4Aa1, Cry4Ba1, and Cry11Aa1. Previous studies with Cyt1Aa1 showed that the LC_50_ and LC_95_ for *Culex quinquefasciatus* SLAB were 47,370 ng/mL and 155,050 ng/mL, respectively; for *Aedes aegypti* were 4219 ng/mL and 22,765 ng/mL, respectively; for *Anopheles gambiae* were 46,557 ng/mL and 129,979 ng/mL, respectively; and for *Anopheles stephensi* were 7780 ng/mL and 13,772 ng/mL, respectively [282]. In contrast, individual Cry toxins against *Culex quinquefasciatus*, for example, are significantly < 500 ng/mL, with Cry 11Aa1 (previously CryIVD) being the most toxic, i.e., 86 ng/mL (LC_50_) and 93 (LC_95_) ng/mL [283]. Nevertheless, Cyt1Aa1 is indispensable to the robust larvicidal activity of Bti. Cyt1Aa1 synergistically interacts with these three Cry proteins against a wide range of mosquito and blackfly species, amplifying their larvicidal activities while delaying and preventing the development of resistance to these toxins [283,284,285], presumably by functioning as a surrogate receptor for these toxins [230,286,287].

The synergistic effect of Cyt1Aa1 extends to Ls’s Tpp1Aa1/Tpp2Aa1 against many mosquito species, including *Aedes aegypti*, which is normally refractive to the binary toxin; the cytotoxin also restores the toxicity of *Culex quinquefasciatus*, which developed resistance to Ls [230,288,289,290]. Interestingly, as *Aedes aegypti* lacks a GPI-anchored amylomaltase ortholog receptive to the Tpp2Aa1 ligand, and as the binary toxin elicits its effect intracellularly, exactly how Cyt1Aa1 facilitates the activity of the binary toxin or the toxin domain, Tpp1Aa1, in this species remains to be resolved. Translocation of the binary toxin into the cytoplasm mediated through irregular membrane perturbations where Cyt1Aa1 interacts with molecules bound to, but not inserted, into the membrane (“detergent-like model”), or forming membrane pores of 6–20 angstroms in diameter (“pore-forming model”), have been suggested [291,292,293,294,295,296]. More recent structural insights by Tetreau et al. [271] opposed the pore-forming and detergent-like models and suggested that interactions of non-porous Cyt1Aa1 oligomers, called membrane-bound aggregates MBA and MBA protomers, can lead to the formation of large pores of >54 nm in diameter that can accommodate the translocation of large molecules, such as the 42 kDa Tpp1Aa1, or perhaps Tpp1Aa1/Tpp2Aa1.

#### 5.2.5. Prospects for Engineering More Robust Strains of Bti

As mentioned above, a unique strategy has evolved in Bti to package and deliver its Cry4Aa1, Cry4Ba1, Cry11Aa1, and Cyt1Aa1 larvicidal toxins as a single composite unit, i.e., the prokaryotic insect larvicidal organelle, PILO [127]. This is the essential reason why commercial formulations of the natural strain of Bti are successful and robust in the field. In particular, the “built-in” Cyt1Aa1-based synergism strategy integral to the PILO is key to preventing or delaying the development of resistance to the Cry proteins in natural populations of vector mosquitoes. Indeed, resistance to Bti in *Aedes*, *Anopheles*, and *Culex* mosquitoes has not been reported, even after 40 years of use. With regard to *Aedes aegypti*, the long-term efficacy of Bti has even been demonstrated under laboratory conditions where this prolific vector was exposed to 0.5 mg/L of VectoBac throughout 30 generations, and where the offspring of survivors remained susceptible to Bti’s composite PILO, with LC_50_ and LC_95_ ranging from 0.013 to 0.022 mg/L and from 0.030 to 0.049 mg/L, respectively [297].

As Bti remains a safe and highly efficacious biopesticide, there have been few innovations to warrant the release of a competitive engineered strain for applied purposes. Regardless, a number of studies have been conducted to determine the molecular mechanisms responsible for the expression and synthesis of larvicidal component systems in Bti [221], but the knowledge gained from such studies has rarely been used to manipulate Bti for practical commercial purposes. To date, there has only been one report clearly demonstrating that a recombinant strain of Bti IPS-82 can be impressively more toxic than parental Bti or Ls 2362, with 21-fold and 32-fold more toxicity, respectively [298]. The Bti IPS-82 recombinant was manipulated to express its native composite PILO, and the Ls Tpp1Aa1/Tpp2Aa1 operon under control of strong sporulation-dependent promoters of Cyt1Aa1 [299] and the mRNA stabilizing sequence (STAB-SD) of Cry3A [300,301]. The engineered strain produced ~472 g/mL of Cry4Aa1, Cry4Ba1, Cry11Aa1, Cyt1Aa1, and Tpp1Aa1/Tpp2Aa1 compared to 250 g/mL for parental Bti-IPS82 (190% increase), and over 450% more Tpp1Aa1/Tpp2Aa1 than Ls 2362. Park et al. [302] also engineered Bti IPS-82 to produce the components of its PILO and Cry11B, but the Bti IPS-82/Cry11B recombinant was only twice as toxic as the parental strain.

Other approaches have been attempted to develop more commercially competitive Bti strains using genes that code for non-larvicidal toxins, including chitinases. The intent of the latter is that the enzyme, once released in the midgut, could weaken the chitin-containing peritrophic membrane, thereby decreasing the time required while concomitantly increasing access for activated toxins to bind to membrane receptors to induce larval lethality. Toward this end, Juarez-Hernandez et al. [303] engineered Bti to produce its PILO and inclusions of a chitinase, ChiA74. Despite developing a method to produce normally soluble ChiA74 as stable inclusions in Bti during sporulation, and demonstrating that the enzyme was indeed active at pH 6–8, the recombinant was only twice as toxic to *Aedes aegypti* compared to parental Bti (LC_50_ 9 ng/mL versus 19.8 ng/mL). Whether or not the narrow pH range for ChiA74 limited its intended effect in the alkaline larval midgut is not clear. Nonetheless, the study shed light on the fact that additional toxins, enzymes, or other biomolecules produced in Bti during sporulation should ideally be particulate; if soluble, they will likely be eliminated during the production and commercial formulation process. Moreover, these molecular larvicidal additives must be easily solubilized and activated in the alkaline environment of the mosquito larval midgut.

Whether current or future engineered strains will be as toxic to mosquito larvae as the Bti IPS-82 that produces its native PILO and Tpp1Aa1/Tpp2Aa1 [298] remains to be seen. Even if such strains were engineered, their applied use may be overshadowed by overriding industry interests, including costs related to the fermentation and formulation process and the marketing of a new product. Finally, as more insights are gained into the structural characteristics of Bti’s PILO, other synthetic biology strategies could evolve in which targeting of biomolecules, including enzymes, such as chitinases, and mosquitocidal toxins, such as Cry11B, through the MFM pores into the lumen of the PILO could result in more robust Bti strains for applied use [127].

## 6. Conclusions

Despite successes in integrated pest management programs to combat mosquitoes, arboviral diseases, such as dengue and dengue hemorrhagic fever, yellow fever, chikungunya, West Nile, and Zika, and parasitic diseases, such as malaria, lymphatic filariasis, and river blindness, continue to threaten the health and well-being of half the world’s population. Though less conspicuous, the threat also applies to feral and domesticated animals, the latter of which are of economic concern. The perpetual problem inflicted by vector-borne diseases, compounded by the development of resistance to synthetic pesticides, globalization, and climate change, which is the most significant factor implicated in the geographic range expansion of mosquitoes, cannot be ignored. Whereas the use of synthetic chemical pesticides will continue to be a core component of vector control and IPMPs in the foreseeable future, the rapidly growing trend to advance “green” eco-friendly technologies to mitigate the perpetual mosquito threat is encouraging but still requires objective oversight by experts in various disciplines, and the public at large. As SIT, IIT, RIDL, and CRISPR/Cas9/Cas13 gene drive systems are refined, and the widespread use of biocontrol agents, such as *Lysinibacillus sphaericus* and *Bacillus thuringiensis* subsp. *Israelensis* (Bti), continues, the future of mosquito control may not be as daunting as it has been in previous decades and, for that matter, centuries.

## Figures and Tables

**Figure 1 biology-13-00182-f001:**
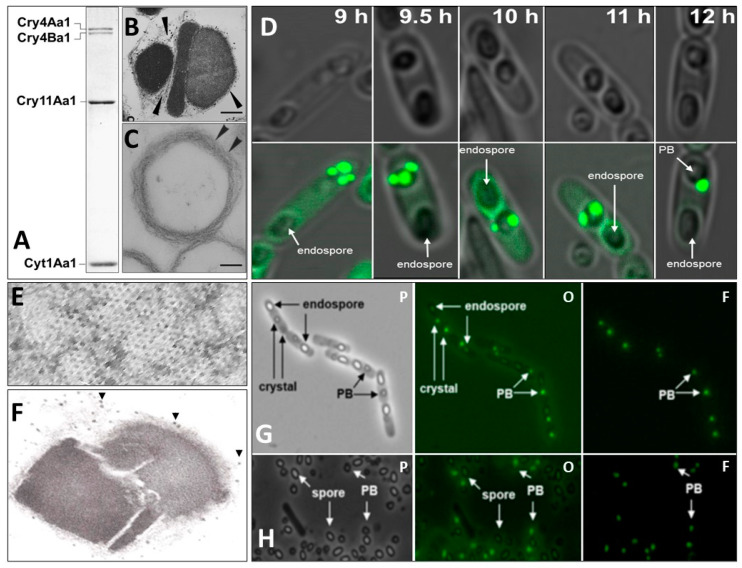
Structural features of *Bacillus thuringiensis* subsp. *israelensis* parasporal body (PB), a unique prokaryotic insect larvicidal organelle (PILO). Sodium dodecyl polyacrylamide gel electrophoresis protein profile showing that Cry4Aa1 (135 kDa), Cry4Ba1 (128 kDa), Cry11Aa1 (65 kDa), and Cyt1Aa1 (27 kDa) (**A**) are the major mosquito larvicidal proteins of the PILO (**B**). Note that crystals of Cry4Aa1/Cry4Ba1, Cry11Aa1, and Cyt1Aa1 are each enveloped by a multilamellar fibrous matrix (MFM; arrowheads) that is also found in the peripheral composite structure. Purified MFM derived from the PILO treated with alkaline to dissolve and remove the Cry and Cyt proteins; multiple layers of the MFM are observed (arrowheads) (**C**). Brightfield and corresponding fluorescence confocal microscopy showing the progressive formation of Bti’s PILO; GFP-labeled Bt0152, a pBtoxis-coded protein, known to specifically bind to the MFM [266], associates with this structure as early as 9 h, well before the three crystalline inclusions are observed at 12–20 h (**D**). Ultrastructural analysis of the MFM showing that it contains hexagonal pores (**E**) and discrete particles thought to be ribosomes (arrowheads) attached to the MFM (**F**), suggesting that the Cry and Cyt proteins are synthesized through the pores and are concentrated and crystallized in their respective compartment in the PILO. GFP-labeled Bt075, a protein also coded for by pBtoxis and which is structurally similar to phage capsid and encapsulin shell proteins, is also a PILO-specific component that associates with the MFM as early as 6 h before distinct crystals are observed; phase (p), phase-fluorescence overlay (o), and fluorescence (f) in sporulating cells at 20 h (**G**); and cells that have autolyzed at 48 (**H**) to release spores (s) and the PILO are shown. Bar (**B**,**C**,**F**) = 0.2 mm. (Adapted from Rudd et al. [127]).

## Data Availability

Not applicable.

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
