# Peer review of "The Perpetual Vector Mosquito Threat and Its Eco-Friendly Nemeses"

_biology, 2024, doi:10.3390/biology13030182_

Round 1

Reviewer 1 Report

Comments and Suggestions for Authors

The manuscript is very well written and brings interesting information about the health issues imposed by vector mosquitoes, traditional vector control strategies that employ synthetic pesticides, and “green” eco-friendly technologies that include SIT, IIT, RIDL, CRISPR/Cas9/Cas13 gene drive systems, and biological control, with emphasis on Lysinibacillus sphaericus and Bacillus thuringiensis subsp. israelensis (Bti).

The authors cite in the summary and conclusion that integrated pest management programs are successful, but I don’t agree with this declaration. If the programs succeeded, the diseases would no longer be a huge problem. Maybe the authors want to say that some programs have success, but they are not the majority.

Lines 32-33: “Integrated pest management programs (IPMPs) have achieved considerable success in mitigating regional transmission”. I think it would be interesting if the authors could cite some examples.

The authors use the phrase "development of resistance to pesticides/insecticides” in different parts of the text. I think the word “development” should be used with caution and, in many parts, replaced by the word “selection”. As we know, insecticide resistance results from random mutations that occur in a given population and are selected by the constant use of insecticides. So, it results from natural selection, and the pesticide is the driving force.

Topic 3: as the authors point out in the text, IPMP uses several vector control methods that have demonstrated effectiveness when used alone or in combination. However, IPMP is not only focused on control strategies. It is conceived as a flexible management system that follows cyclical processes with multiple rounds of situational analysis, planning, design, implementation, monitoring and evaluation (OPAS, 2019). Considering only control methods, IPMP involves mechanical, environmental, biological, chemical, and educational approaches, so I don't think that the authors should highlight only the strategies based on synthetic pesticides. I think they should also include a brief description of the other methods.

Line 245: the correct is “the stage that transmits pathogens to…”; the vectors transmit the etiological agent, and whether it will develop in disease is a different story and includes other factors.

Topic 3.1: the authors cite the main classes of insecticides traditionally used for vector control, but we can’t forget that other classes have also been in use for the past years and deserve to be mentioned, like neonicotinoids, spinosyns, pyrroles and IGR.

Line 273: I don’t understand why the authors cited only one paper about resistance to temephos in India. Resistance to temephos (as well as to all other classes of insecticides) has been observed in several countries, not just India. The authors should consider citing this interesting and important review published by Moyes et al, 2017 about the status of insecticide resistance in the major Aedes vectors of arboviruses. The paper provides evidence for the geographical distribution of insecticide resistance in two major vectors worldwide and maps the data collected for the four main classes of neurotoxic insecticide (carbamates, organochlorines, organophosphates, and pyrethroids). It also brings discussion about the mechanisms involved.

Lines 289-300: I don't understand the relation between this explanation about the GABA system and the resistance mechanism of target site alteration. I think this part should be deleted. Instead, the authors could include information and publications about mutations, vectors and countries related to the cited mechanism.

Lines 606-632: The information provided in this part is a copy of box 1. There is no need for duplication, so box 1 can be deleted.

Minor corrections

Line 436: the correct spelling is JUAZEIRO.

Line 491: the correct is “cycle IS also promising”.

Line 729: punctuation is missing.

Line 746: the paragraph has a break.

Reference cited:

Organização Pan-Americana da Saúde. Documento operacional para a execução do manejo integrado de vetores adaptado ao contexto das Américas. Washington, D.C.: OPAS; 2019.

Moyes CL, Vontas J, Martins AJ, Ng LC, Koou SY, Dusfour I, et al. (2017) Contemporary status of insecticide resistance in the major Aedes vectors of arboviruses infecting humans. PLoS Negl Trop Dis 11(7): e0005625. https://doi.org/10.1371/journal.pntd.0005625

Author Response

We would like to express our sincere appreciation to the REVIEWER for the time taken to review our submission and to provide insightful comments and suggestions that led to a more refined manuscript.  Below, we provide responses to the Reviewer's comments:

REVIEWER’S comment: The manuscript is very well written and brings interesting information about the health issues imposed by vector mosquitoes, traditional vector control strategies that employ synthetic pesticides, and “green” eco-friendly technologies that include SIT, IIT, RIDL, CRISPR/Cas9/Cas13 gene drive systems, and biological control, with emphasis on Lysinibacillus sphaericus and Bacillus thuringiensis subsp. israelensis (Bti).

Authors’ response: We thank the Reviewer for this comment.

REVIEWER’S comment: The authors cite in the summary and conclusion that integrated pest management programs are successful, but I don’t agree with this declaration. If the programs succeeded, the diseases would no longer be a huge problem. Maybe the authors want to say that some programs have success, but they are not the majority.

Authors’ response: Indeed, the Reviewer is correct. In the strictest sense regarding its application, IPMPs have not been successful, as the proliferation of mosquitoes and vector-borne diseases are still pervasive. As such, we have removed the term and replaced it with “some success” and additional comment.  (see lines 24).

REVIEWER’S comment: Lines 32-33: “Integrated pest management programs (IPMPs) have achieved considerable success in mitigating regional transmission”. I think it would be interesting if the authors could cite some examples.

Authors’ response: Indeed, the Reviewer is correct. As above, revisions have made in the manuscript. (see lines 24, 38-40).

“Integrated pest management programs (IPMPs) have achieved some success in mitigating regional transmission and persistence of these diseases. However, as many vector-borne diseases remain pervasive, it is obvious that IPMP successes have not been absolute in eradicating the threat imposed by mosquitoes.”

REVIEWER’S comment: The authors use the phrase "development of resistance to pesticides/insecticides” in different parts of the text. I think the word “development” should be used with caution and, in many parts, replaced by the word “selection”. As we know, insecticide resistance results from random mutations that occur in a given population and are selected by the constant use of insecticides. So, it results from natural selection, and the pesticide is the driving force.

Authors’ response: The Reviewer is correct. The text has been revised. (see lines 306-311)

These chemicals can directly harm non-target invertebrate and vertebrate species, accumulate in the environment, and affect food webs for protracted periods, and in particular, impose selective pressures leading to the persistence of resistant mosquito populations [83,84,85]. Regarding the latter, resistance to organophosphates, including larvicidal temephos and chlorpyrifos, carbamates, organochlorines, and pyrethroids has been documented for species of Aedes, Culex, and Anopheles [86,87,88,89,90].

REVIEWER’S comment: Topic 3: as the authors point out in the text, IPMP uses several vector control methods that have demonstrated effectiveness when used alone or in combination. However, IPMP is not only focused on control strategies. It is conceived as a flexible management system that follows cyclical processes with multiple rounds of situational analysis, planning, design, implementation, monitoring and evaluation (OPAS, 2019). Considering only control methods, IPMP involves mechanical, environmental, biological, chemical, and educational approaches, so I don't think that the authors should highlight only the strategies based on synthetic pesticides. I think they should also include a brief description of the other methods.

Authors’ response: The Reviewer is correct. We have revised the manuscript accordingly. (see lines 251-264)

“Effective mosquito abatement and disease prevention strategies employ integrated approaches, and include at least seven components: (1) mosquito surveillance in which the use of various types of traps are useful for cataloguing the vector species that are present in a geographic region, (2) physical mosquito control or source reductions focused on eliminating mosquito land and aquatic breeding sites, (3) mosquito larval control measures using chemical or biological control, (4) adult mosquito control (aduticiding) using chemical pesticides that target the stage the transmits viral and parasitic agents of human and animal disease, (5) insect resistance monitoring using cage trials and bioassays – a component that cannot be ignored as mosquitoes are extremely adaptable and can have multiple generations in a single transmission season, (6) public education on measures that can be taken to reduce or eliminate potential mosquito breeding sites, how to avoid mosquito bites, and clinical symptoms associated with vector-borne diseases, and (7) accurate record keeping to establish year-to-year trends and breeding sites, and for regulatory compliance [55,56,57,58].”

REVIEWER’S comment: Line 245: the correct is “the stage that transmits pathogens to…”; the vectors transmit the etiological agent, and whether it will develop in disease is a different story and includes other factors.

Authors’ response: The Reviewer is correct. We have revised the manuscript accordingly (see line 257)

“…using chemical pesticides that target the stage the transmits viral and parasitic agents of human…”

REVIEWER’S comment: Topic 3.1: the authors cite the main classes of insecticides traditionally used for vector control, but we can’t forget that other classes have also been in use for the past years and deserve to be mentioned, like neonicotinoids, spinosyns, pyrroles and IGR.

Authors’ response: The Reviewer is correct. We have revised the manuscript accordingly. Two paragraphs were added to address this comment. (see lines 281-303).

“Apart from the more traditional pesticides, others used in recent years include neonicotinoids, spinosyns, pyrroles, and insect growth regulators (IGR). Neonicotinoids, like nicotine, bind to nicotinic ACh receptors (nAChRs), and include dinotefuran and clothianidin which show promise in mosquito control [69,70]. Under normal physiological circumstances low to moderate activation of nAChRs by ACh elicit nervous stimulation, whereas high levels of the neurotransmitter overstimulate and block these receptors, resulting in paralysis and death. Unlike ACh that is broken down by acetylcholinesterase, the enzyme has no effect on neonicotinoids. The pesticide binds irreversibly to the enzyme leading to paralytic death of the insect [71]. Spinosyns are metabolites produced by the soil bacterium, Saccharopolyspora spinosa. Members of the spinosyn family of insecticides, including Spinosad, which is composed of spinosyns A and D, have a unique mode action in that they disrupt AChR in a wide variety of arthropods, including mosquitoes, particularly Aedes and Culex [72,73,74].

                Pyrroles and IGRs are viable alternatives to neurotoxins in mosquito control. Pyrroles, including chlorofenapyr, are broad spectrum insecticides, that unlike neurotoxins, disrupt respiratory pathways and proton gradients through uncoupling of oxidative phosphorylation in the mitochondria, and are effective in bed net and indoor treatments for the control of Anopheles, Culex, and pyrethroid-resistant Aedes aegypti [76,78,77,79]. The use of IGRs in IPMPs are attractive because of their low toxicity to mammals and non-target species. IGRs elaborate their effect by disrupting insect development; for example, methoprene mimics juvenile hormones and prevents larvae from completing their immature stage thereby reducing the adult population, and pyriproxyfen inhibits chitin synthesis which is essential for formation of the exoskeleton of insects [80,81,82].”

REVIEWER’S comment: Line 273: I don’t understand why the authors cited only one paper about resistance to temephos in India. Resistance to temephos (as well as to all other classes of insecticides) has been observed in several countries, not just India. The authors should consider citing this interesting and important review published by Moyes et al, 2017 about the status of insecticide resistance in the major Aedes vectors of arboviruses. The paper provides evidence for the geographical distribution of insecticide resistance in two major vectors worldwide and maps the data collected for the four main classes of neurotoxic insecticide (carbamates, organochlorines, organophosphates, and pyrethroids). It also brings discussion about the mechanisms involved.

Authors’ response: The Reviewer is correct. We have revised the manuscript accordingly. (see lines 304-312.

“Synthetic pesticides are generally quite effective mosquito adulticides. Despite the rapid kill they induce, unintended negative environmental and ecological impacts cannot be ignored. These chemicals can directly harm non-target invertebrate and vertebrate species, accumulate in the environment, and affect food webs for protracted periods, and in particular, impose selective pressures leading to the persistence of resistant mosquito populations [83,84,85]. Regarding the latter, resistance to organophosphates, including larvicidal temephos and chlorpyrifos, carbamates, organochlorines, and pyrethroids has been documented for species of Aedes, Culex, and Anopheles [86,87,88,89,90]. Taken together and apart from climate change and globalization, resistance to synthetic insecticides is a major contributor to the proliferation of mosquitoes and spread of infectious diseases globally.”

REVIEWER’S comment: Lines 289-300: I don't understand the relation between this explanation about the GABA system and the resistance mechanism of target site alteration. I think this part should be deleted. Instead, the authors could include information and publications about mutations, vectors and countries related to the cited mechanism.

Authors’ response: Indeed, the Reviewer highlighted an important deficiency. In retrospect, we should have been more careful in explaining the significance of the GABAergic system.

We have revised the manuscript accordingly, to include applicable mutations, and have separated this from the GABAergic system that apparently is associated with pathogen propagation in mosquito hosts. (see lines 325-359)

  “Regarding insecticide target site specificity, certain site-specific mutations in sodium channel proteins, acetylcholinesterases, and GABA receptors are strongly associated with resistance to their ligand pesticides. For example, Studies by Xu et al. [96] and Li et al. [97] on the sodium channel of Culex quinquefasciatus showed that at least three specific nonsynonymous mutations (A109S, L982F, W1573R) were directly associated with resistance to permethrin and that six synonymous mutations (codons for L582, G891, A241, P1249, G1733) that do not alter the amino acid sequence may play a role in the evolution of resistance. In Culex and Anopheles, in addition to other insect species that display insensitivity or reduced sensitivity to organophosphates and carbamates, a mutation in the ache1 gene conferring a G119S substitution likely causes steric hindrance that reduces the accessibility of the inhibitor pesticide substrate to acetylcholine esterase 1, AChE1 [83,98,99,1100,101]. The major neuronal inhibitory mechanism in insects (and vertebrates) is the GABAergic system, in which activation suppresses neuronal excitability. The GABA receptor is targeted by dieldrin and fipronil, respectively, cyclodiene and phenyl pyrazole insecticides. Mutations resulting A296S/G substitutions in the GABA receptor is associated with dieldrin resistance in many insects, including Anopheles gambiae (A296G), and Anopheles arabiensis, Anopheles stephensi, Anopheles funestus, and Aedes aegypti (A296S), and generally lower levels of resistance to fipronil [83,102,103,104,105,106,107].

     In other regards, it is interesting to note that the GABAergic system also plays an important role in immune regulation in mammals, for example, in autoimmune inflammation and migration of immune cells in response to parasitic infection with Toxoplasma gondii [108,109,110]. It is now apparent that a similar role for GABA signaling occurs in mosquitoes. Zhu et al. [111] showed that (i) dsRNA-mediated disruption of GABA and the specific inhibition of GABAA receptor decrease arboviral replication, whereas the introduction of glutamic acid per os increases the ability of arboviruses to infect mosquitoes, (ii) blood meals enhance viral replication through GABAergic activation, and (iii) the GABAergic system suppresses the Imd pathway, a NF-kB pathway known to regulate bacterial and malarial infection in mosquitoes [112,113,114]. Given this scenario, the extent to which sublethal levels of insecticides dampen or inhibit the GABAergic system and how resistance to these synthetics influence the propagation of pathogens in natural mosquitoes populations remains to be resolved. Nevertheless, Zhu et al. [111] demonstrated that at least two GABA inhibitors, fipronil and bilobalide, markedly reduced dengue (DENV-2) and Zika virus loads in Aedes aegypti that survived treatment with these chemicals, a finding that suggests that inhibitors of the GABAergic system may play a role in reducing the dissemination of arbovirus in the field.”

REVIEWER’S comment: Lines 606-632: The information provided in this part is a copy of box 1. There is no need for duplication, so box 1 can be deleted.

Authors’ response: The Reviewer is correct. We removed “BOX 1” from the manuscript.

REVIEWER’S comments: Minor corrections

Line 436: the correct spelling is JUAZEIRO. Authors’s response: Corrected in the revised version.

Line 491: the correct is “cycle IS also promising”. Authors’s response: Corrected in the revised version.

Line 729: punctuation is missing. Authors’s response: Corrected in the revised version.

Line 746: the paragraph has a break. Authors’s response: Corrected in the revised version.

REVIEWER’S suggestion: Reference cited:

Organização Pan-Americana da Saúde. Documento operacional para a execução do manejo integrado de vetores adaptado ao contexto das Américas. Washington, D.C.: OPAS; 2019.

Moyes CL, Vontas J, Martins AJ, Ng LC, Koou SY, Dusfour I, et al. (2017) Contemporary status of insecticide resistance in the major Aedes vectors of arboviruses infecting humans. PLoS Negl Trop Dis 11(7): e0005625. https://doi.org/10.1371/journal.pntd.0005625

Authors’ response: These references are now cited in the manuscript. See references 56 and 87.

Again, we thank the REVIEWER for the insightful comments and suggestions that led to a more refined manuscript.

Reviewer 2 Report

Comments and Suggestions for Authors

Nicely written paper for its length. Interesting to read. A few suggested changes for clarity here and there. See file.

Insect names, in American English, that contain the word fly are one word only if the insect is not actually a fly, e.g., butterfly. Black fly is a dipteran, thus it is 2 words.

Aedes aegypti is used 50 times in the manuscript, but the naming authority, Linnaeus, is not placed after the species name until line 640. Usually, the naming authority is added at and only at the first mention of the genus and species. This should be done for all species names in this paper.

The Negev constitutes most of southern Israel west of the Jordan river. It includes the Dead Sea down to the Red Sea, which is in the south.

Line 657-58: use NTD only. No need to repeat the name.

Author Response

We would like to express our sincere appreciation to the REVIEWER for the time taken to review our submission and to provide insightful comments and suggestions that led to a more refined manuscript.  Below, we provide responses to the Reviewer's comments:

Comments and Suggestions for Authors

REVIEWER’S comment: Nicely written paper for its length. Interesting to read. A few suggested changes for clarity here and there.

Authors’ response: We thank the Reviewer for this comment.

Insect names, in American English, that contain the word fly are one word only if the insect is not actually a fly, e.g., butterfly. Black fly is a dipteran, thus it is 2 words.

Authors’ response: The Reviewer is correct. The changes were made in the revised manuscript.  (see line 60). “black flies, tsetse flies, sand flies”

Aedes aegypti is used 50 times in the manuscript, but the naming authority, Linnaeus, is not placed after the species name until line 640. Usually, the naming authority is added at and only at the first mention of the genus and species. This should be done for all species names in this paper.

Authors’ response: The Reviewer is correct. The changes were made in the revised manuscript. (see lines 179 and 190)

“Culex. p. quinquefasciatus (Linnaeus); Aedes aegypti (Linnaeus)

The Negev constitutes most of southern Israel west of the Jordan river. It includes the Dead Sea down to the Red Sea, which is in the south.

Authors’ response: The Reviewer is correct. The description was changed accordingly. (see line 691-693).

“The bacterium was isolated from dead Culex pipiens larvae found in a stagnant pond in the north central Negev Desert and it was shown to be lethal to larvae of Nematoceran insects (e.g., mosquitoes, blackflies, and chironomid midges) [249].”

Line 657-58: use NTD only. No need to repeat the name

Authors’ response: The Reviewer is correct. We used the term “NTD” after its initial designation.

“neglected tropical disease, NTD [9,10].” See line 94, and thereafter, NTD (lines 99, 133, 713)

Again, we thank the REVIEWER for the insightful comments and suggestions that led to a more refined manuscript.